# Tumor purity-related genes for predicting the prognosis and drug sensitivity of DLBCL patients

Zhenbang Ye[1†], Ning Huang[2†], Yongliang Fu[1], Rongle Tian[1], Liming Wang[2*], Wenting Huang[1,3*]

[1]Department of Pathology, National Cancer Center/National Clinical Research Center for Cancer/Cancer Hospital, Chinese Academy of Medical Sciences and Peking Union Medical College, Beijing, China; [2]Department of Hepatobiliary Surgery, National Cancer Center/National Clinical Research Center for Cancer/Cancer Hospital, Chinese Academy of Medical Sciences and Peking Union Medical College, Beijing, China; [3]Department of Pathology, National Cancer Center/National Clinical Research Center for Cancer/Cancer Hospital & Shenzhen Hospital, Chinese Academy of Medical Sciences and Peking Union Medical College, Shenzhen, China

*For correspondence:
stewen_wang@sina.com (LW);
huangwt@cicams.ac.cn (WH)

[†]These authors contributed equally to this work

## Abstract

**Background:** Diffuse large B-cell lymphoma (DLBCL) is the predominant type of malignant B-cell lymphoma. Although various treatments have been developed, the limited efficacy calls for more and further exploration of its characteristics.

**Methods:** Datasets from the Gene Expression Omnibus (GEO) database were used for identifying the tumor purity of DLBCL. Survival analysis was employed for analyzing the prognosis of DLBCL patients. Immunohistochemistry was conducted to detect the important factors that influenced the prognosis. Drug-sensitive prediction was performed to evaluate the value of the model.

**Results:** VCAN, CD3G, and C1QB were identified as three key genes that impacted the outcome of DLBCL patients both in GEO datasets and samples from our center. Among them, VCAN and CD3G+ T cells were correlated with favorable prognosis, and C1QB was correlated with worse prognosis. The ratio of CD68 + macrophages and CD8 + T cells was associated with better prognosis. In addition, CD3G+T cells ratio was significantly correlated with CD68 + macrophages, CD4 + T cells, and CD8 +T cells ratio, indicating it could play an important role in the anti-tumor immunity in DLBCL. The riskScore model constructed based on the RNASeq data of VCAN, C1QB, and CD3G work well in predicting the prognosis and drug sensitivity.

**Conclusions:** VCAN, CD3G, and C1QB were three key genes that influenced the tumor purity of DLBCL, and could also exert certain impact on drug sensitivity and prognosis of DLBCL patients.

**Funding:** This work is supported by the Shenzhen High-level Hospital Construction Fund and CAMS Innovation Fund for Medical Sciences (CIFMS) (2022-I2M-C&T-B-062).

## eLife assessment

This study presents a predictive scoring system in DLBCL based on the expression of three tumour microenvironment-related genes. Such a scoring system seems **useful** for predicting tumour purity levels in DLBCL. The provided evidence showing an association between worse DLBLC prognosis and high-risk score is **solid**, but it is **incomplete** to draw a clear conclusion about the links between risk score and drug sensitivity.

## Introduction

The latest refined classification by the World Health Organization (WHO) categorizes large B-cell lymphoma as a heterogeneous group of B-cell lymphomas (*Alaggio et al., 2022*). DLBCL is the most prevalent type among them, accounting for around 30% of all non-Hodgkin lymphomas. DLBCL was initially classified by the Hans algorithm into two subtypes: germinal center B-cell-like (GCB) and non-GCB (*Hans et al., 2004*). Further differentiation has since identified three distinct gene-based subtypes: GCB, activated B-cell-like (ABC), and the unclassified category (*Sehn and Salles, 2021*). After undergoing R-CHOP chemotherapy, about 60% of patients achieve long-term remission; however, approximately 30% of patients experience relapse, resulting in poor prognosis and a considerable number of deaths from refractory lymphoma (*Autio et al., 2021*). Consequently, to explore the characteristics of DLBCL in detail is urgently needed for developing more effective therapy.

Solid tumor tissue comprises tumor cells and the surrounding stroma, which encompasses diverse types of matrix cells, immune cells, endothelial cells (*Joyce and Pollard, 2009*), etc. The tumor micro-environment (TME) is a complex and dynamic system that consists of the extracellular matrix and a variety of cellular components. Recent studies have unveiled multiple subgroups of immune cells within the microenvironment of DLBCL, including T cells, B cells, NK cells, monocytes/macrophages, dendritic cells, as well as the distribution of stromal cell components like fibroblasts and endothelial cells (*Steen et al., 2021*; *Ciavarella et al., 2018*). Despite the relatively limited composition of the TME in DLBCL, its role in tumor proliferation and evasion of the immune system should not be disregarded. The interaction between tumors and the microenvironment is a vital factor that impacts the development and prognosis of B-cell lymphoma (*Ennishi et al., 2020*). Nevertheless, the existing research on the influence of the TME on the prognosis of DLBCL patients is limited and lacks a consensus.

Moreover, the comprehensive investigation of non-immune cell components in the TME is still lacking. Previous research on stroma in DLBCL has predominantly indicated that a higher quantity of extracellular matrix is associated with a more favorable prognosis, while increased vascular density is associated with poorer prognosis (*Miyawaki et al., 2022*). Furthermore, higher stromal scores have been associated with an improved prognosis in DLBCL patients (*Schmitz et al., 2018*). Additionally, a fibrotic tumor microenvironment has been correlated with a better prognosis after DLBCL chemotherapy and immunotherapy (*Lou et al., 2022*). These research findings stem from computational analysis of stromal and immune scoring in gene databases and have not been experimentally validated as of yet.

Tumor purity quantifies the relative ratio of tumor cells to the surrounding stromal components in solid tumors, elucidating the dynamics between tumor cells and their microenvironment (*Gong et al., 2020*). It can partly reflect the characteristics of TME, namely, a higher tumor purity indicates a lower abundance of stromal components in TME. Tumor purity is associated with patients' prognosis, and the strength of this association varies across different tumor types (*Lou et al., 2021*; *Zhao et al., 2023*; *Zhang et al., 2017*). Therefore, when investigating the influence of TME on the prognosis of DLBCL patients, it is crucial to analyze not only the immune cell components but also the significance of non-immune cell components.

This study utilized bioinformatic analysis to establish the relationship between immune and stromal components and the prognostic outcomes of DLBCL patients. We developed a novel immunohisto-chemical panel to assess prognostic outcomes and treatment sensitivity by detecting the expression of VCAN, CD3G, C1QB, CD68, CD4, and CD8 in both the TME and tumor cells of 190 DLBCL patients. We then explored their relationship with DLBCL clinicopathological features as well as overall survival (OS).

## Materials and methods

### Data collection and tumor purity-related genes (TPGs) selection

The RNA-Sequence and clinical data of GSE53786 and GSE32918 datasets were downloaded from GEO database. The first gene symbols of GSE53786 datasets were retained when one probe detected multiple genes. Average expression value of genes in each dataset were calculated and used when one gene was detected by multiple probes. Tumor purity was assessed by ESTIMATE (Estimation of Stromal and Immune cells in Malignant Tumor tissues using Expression data) algorithm (*Yoshihara*

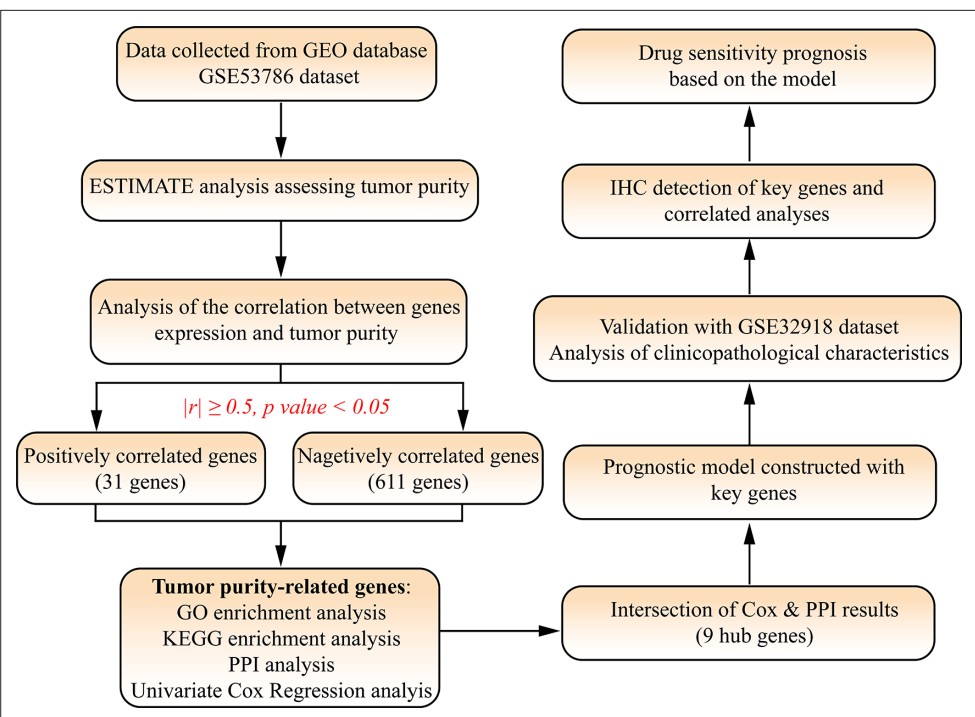

**Figure 1.** Flow chart and study design of this research.

*et al., 2013*) and then its correlation with genes expression was analyzed. The genes with | r |≥0.5 and p-value <0.05 was defined as the TPGs (*Figure 1*).

## TPGs function analysis

Gene Ontology (GO) and Kyoto Encyclopedia of Genes and Genomes (KEGG) analyses were executed to analyze the biological processes, cellular components, molecular functions, and pathways related to the TPGs. The statistical significance was considered as p.adjust <0.05.

The protein-protein interactions (PPI) analysis was utilized to investigate the interaction among TPGs, and those with interactive confidence greater than 0.90 on the STRING platform (version 11.5) were selected to establish an interaction network with Cytoscape software (version 3.8.2).

## Prognostic model

The prognostic model was constructed with 'survival' package in R (version 4.1.3). The genes enrolled in this model was selected among the prognostic and PPI hub TPGs by function 'step' in 'survival' package, which can optimize the model. The prognostic model was represented by riskScore = $\sum_{i=1}^{n}$ gene expression$_i$ × coef$_i$ .

## Clinical specimens and follow-up

190 patients from the Cancer Hospital Chinese Academy of Medical Sciences, the CHCAMS cohort, were enrolled in this study (*Supplementary file 1*). All patients received surgery or biopsy during September, 2010 and September, 2020, and then standard follow-ups were carried out until March, 2023. The overall survival (OS) was defined as the interval between the operation and death or the last follow-up. The specimens from the CHCAMS cohort were used for immunohistochemistry assay.

## IHC

Paraffin-embedded DLBCL tissues of CHCAMS Cohort were used for immunohistochemistry (IHC). After de-paraffinization and hydration, heat-induced method was performed for antigen retrieval. Primary antibody of VCAN (AB177480, 1:100, Abcam, USA), CD3G (AB134096, 1:1000, Abcam, USA), C1QB (AB92508, 1:50, Abcam, USA), CD68 (303565, 1:1000, Abcam, USA), CD4 (ZM-0418, ZSGB-BIO, China), CD8 (ZA-0508, ZSGB-BIO, China), CD206(24595 S, 1:400, CST, USA), and CD32(15625–1-AP,

1:1000, proteintech, China) was incubated at 4 °C overnight. Sections were washed with TBS-T buffer, and then incubated with secondary antibody, and finally stained with DAB. The quantitative analysis of the slices was conducted by QuPath-0.4.3. VCAN and C1QB were assessed by H-score, and CD3G, CD68, CD4, CD8, CD206, CD32 were assessed as the ratio of the corresponding positive cells among all cells.

### Drug sensitivity prediction

Drug sensitivity prediction was conducted utilizing 'oncoPredict' packages in R 4.1.3. The drug sensitivity data was collected from Genomics of Drug Sensitivity in Cancer (GDSC). The drugs that was analyzed in this study was selected according to clinical practice or clinical trials searched in Pubmed.

### Statistical analysis

Data in this study was shown in the form of mean ± SEM. Correlation between two variates was determined with Spearman analysis. Kaplan–Meier (K–M) curve and Log rank test were used for survival analysis. The cut-offs of survival analysis were provided by X-tile. The independent risk factor analysis was performed with Cox regression analysis. Receiver operating characteristic (ROC) curve was used for test the efficacy of prognostic model. The clinicopathological characteristics difference analysis was conducted with $\chi^2$ test, Fisher's exact test, or Wilcoxon rank sum test. The drug sensitivity scores were compared with the Wilcoxon rank sum test. In this study, $p<0.05$ were considered statistically significant.

## Results

### Tumor purity-related genes were correlated with extracellular matrix organization and immune response

Based on GSE53786 dataset, we first assessed the tumor purity of DLBCL, which ranged from 17.2 to 67.4% (*Figure 2A*). In order to screen out the TPGs, we then analyzed the correlation between genes expression and tumor purity. According to the thresholds mentioned above, 642 genes were identified as TPGs, among which 31 genes were positively correlated with tumor purity, while 611 genes were negatively correlated with it (*Figure 2B*). In addition, tumor purity did have influence on the prognosis of DLBCL patients, which showed that patients with high tumor purity had lower OS rates than those with low tumor purity (*Figure 2C*, p=0.025).

Next, we performed GO and KEGG enrichment analyses to explore the functions and signaling pathways in which these TPGs were involved. It turned out that the TPGs were mainly associated with extracellular matrix organization and immune response (*Figure 2D and E*). Not only did the enrichment results confirm that these genes were reliable to be related with the tumor purity, but it also laid solid foundations for the sequent analyses.

### A prognostic model was constructed with three TPGs

With the 642 TPGs, we exerted PPI analysis to investigate their interaction and the hub genes (*Figure 3A*). The TPGs who had five or more interactive genes were shown in *Figure 3B*, and defined as hub genes. Then, we performed univariate Cox regression analysis to figure out the TPGs that were associated with the prognosis of DLBCL patients, and 103 genes were identified (*Figure 3C*). Interestingly, most of the TPGs were correlated with good outcomes (with HR <1), and only six genes were associated with poor outcomes (with HR >1). Through conducting intersection analysis, we found nine genes (LUM, VCAN, YAP1, COL5A2, SDC2, TWIST1, CD3G, C1QB, and C3) were intersection genes (*Figure 3D*), indicating that they had an active effect in modulating the tumor purity, as well as influencing the prognosis of DLBCL patients.

After ascertaining the key genes, we tried to construct a prognostic model with them. The model was constructed by Cox regression, and the three selected genes (VCAN, CD3G, C1QB) and their parameters like coefficient, HR, and 95% CI of HR, were shown in *Figure 4A*. It showed that VCAN and CD3G were correlated with good prognosis and C1QB was correlated with poor prognosis. All patients were divided into high (n = 59) and low-risk (n = 60) groups according to the median value of riskScore (*Figure 4B*). As expected, the high-risk group has worse prognosis than the low-risk group (*Figure 4C*). In addition, the three genes were differentially expressed between high and low-risk

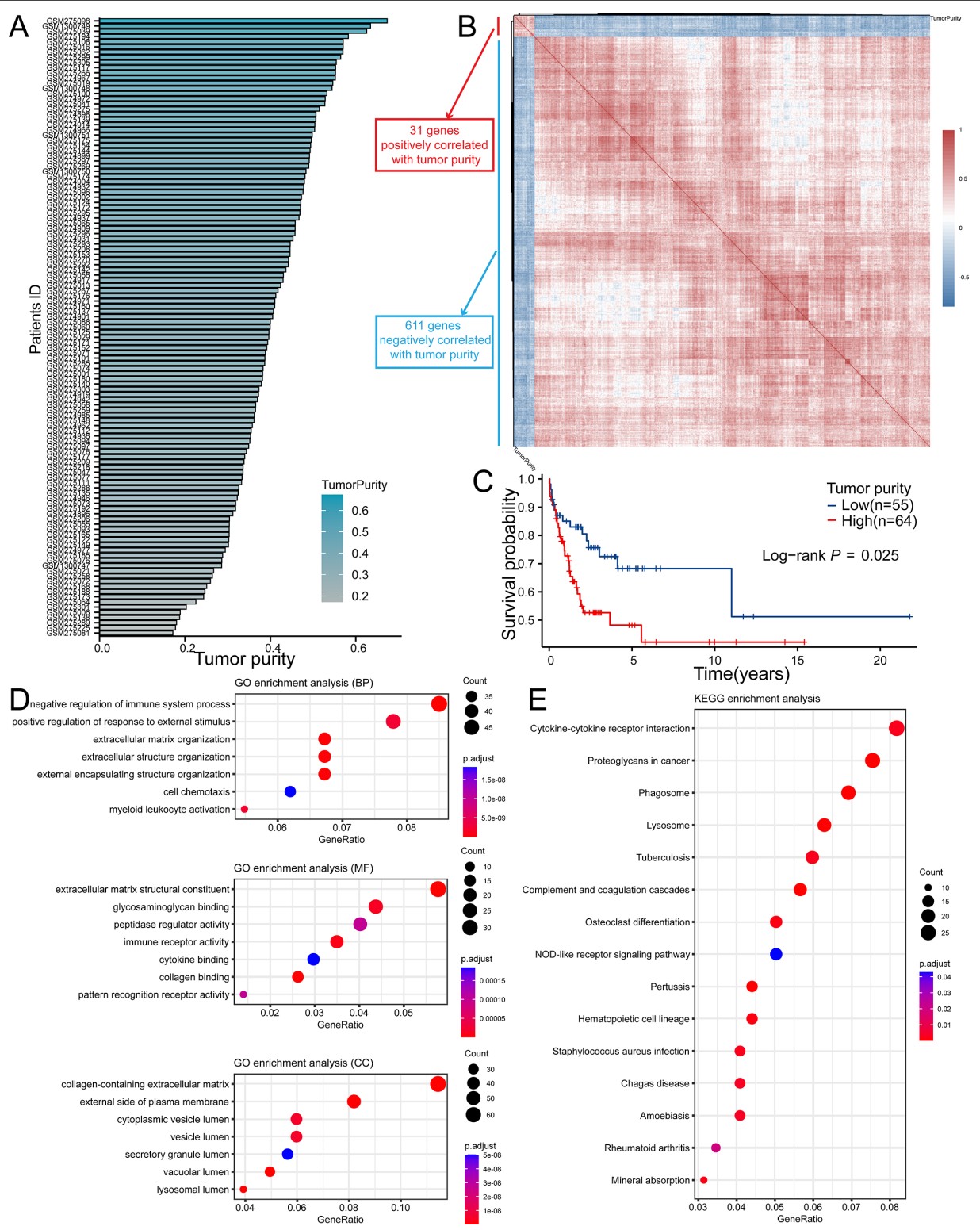

**Figure 2.** TPGs were screened out with GSE53786 dataset. (**A**) The range (17.2–67.4%) of tumor purity of samples in GSE53786. (**B**) The heatmap shows genes defined as TPGs. (**C**) The K–M curve showed high tumor purity was correlated with poor prognosis in DLBCL patients in GSE53678 dataset (Patients were divided into two groups according to the best-cutoff provided by 'survminer' package in R). (**D**) GO analysis of TPGs. (**E**) KEGG analysis of TPGs. TPGs, tumor purity-related genes; K–M, Kaplan-Meier; DLBCL, diffuse large B cell lymphoma; GO, gene ontology; KEGG, Kyoto Encyclopedia of Genes and Genomes.

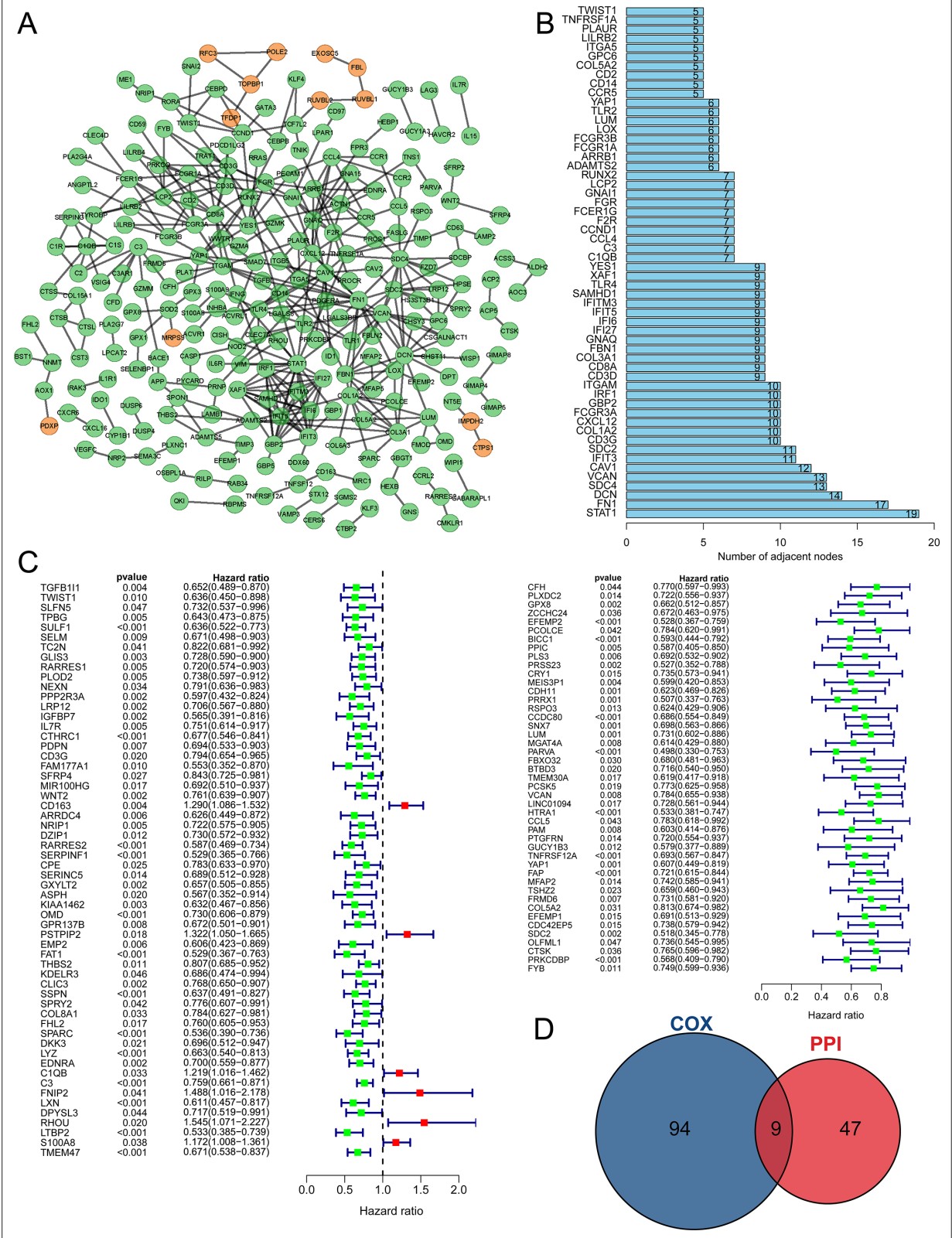

**Figure 3.** The key gene candidates used for constructing prognostic model were selected. (**A**) The PPI network of tumor purity-related genes (TPGs) (orange nodes representing genes positively correlated with tumor purity, and green nodes representing genes negatively correlated with tumor purity). (**B**) The barplot shows hub genes with five or more interactive genes. (**C**) The forest plot showing prognostic TPGs of diffuse large B-cell lymphoma (DLBCL) patients in GSE53786. (**D**) The Venn plot shows intersection genes of PPI hub gene and prognostic TPGs. PPI, protein-protein interaction.

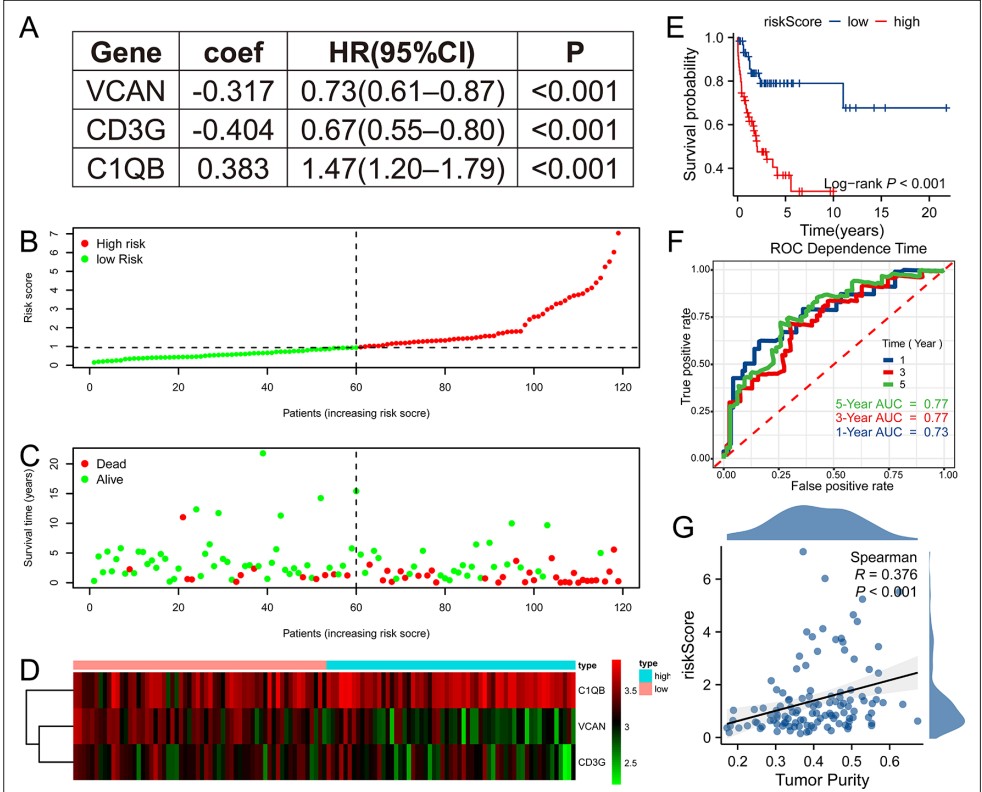

**Figure 4.** Tumor purity-related genes (TPGs) signature prognostic model was constructed. (**A**) Three genes enrolled in the prognostic model. (**B**) The patients in GSE53786 dataset were divided into high (n = 59) and low-risk (n = 60) group according to the median riskScore based on the prognostic model. (**C**) High-risk group had worse prognosis than low-risk group. (**D**) The heatmap shows the expression discrepancy of the three genes. (**E**) Survival analysis revealed that high-risk group had poor prognosis in GSE53786 dataset. (n _low_ = 60, n _high_ = 59). (**F**) The ROC curve showed that the prognostic model performed well in predicting 1 year, 3 year, and 5 year prognosis in GSE53786 dataset. (**G**) Tumor purity was positively correlated with riskScore in GSE53786 dataset. ROC, receiver operating characteristic.

groups, with VCAN and CD3G showing high expression level in low-risk group, and C1QB showing high expression level in high-risk group, which was consistent with the coefficient (*Figure 4D*). To appraise the efficacy of this prognostic model, we conducted survival analysis and ROC analysis. High-risk group had lower OS rate than low-risk group (*Figure 4E*, p<0.001), and the areas under curve (AUC) for 1 year, 3 year, and 5 year ROC were 0.73, 0.77, and 0.77, respectively (*Figure 4F*). Just similar to tumor purity, the high riskScore indicated bad outcome, which was consistent with the positive correlation between tumor purity and riskScore (*Figure 4G*). This TPGs signature prognostic model manifested satisfying prognostic efficacy.

When we applied this model to GSE32918 dataset, it still did excellently and the results were in accordance with that in GSE53786 dataset (*Figures 5A and 4B*; *Figure 5—figure supplement 1*). Next, we analyzed the relationship between riskScore and some clinicopathological characteristics provided in GSE53786 dataset. The results showed that high-risk group had more ABC type DLBCL, while low-risk group had more the GCB type DLBCL (*Figure 5C*). Besides, high-risk group displayed higher lactic dehydrogenase (LDH) ratio (*Figure 5F*). However, the Eastern Cooperative Oncology Group (ECOG) performance and stage was not associated with the riskScore (*Figure 5D and E*). Still, the high-risk group has more Stage III and Stage IV patients, but less Stage I patients than low-risk group. Finally, we employed the univariate and multivariate analysis to explore whether riskScore was an independent prognostic factor for DLBCL patients. As expected, the riskScore was associated with the poor prognosis (*Figure 5G*, p<0.001, HR = 1.545, 95% CI 1.284–1.861) and was an independent prognostic factor (*Figure 5H*, p=0.002, HR = 1.474, 95% CI 1.156–1.879).

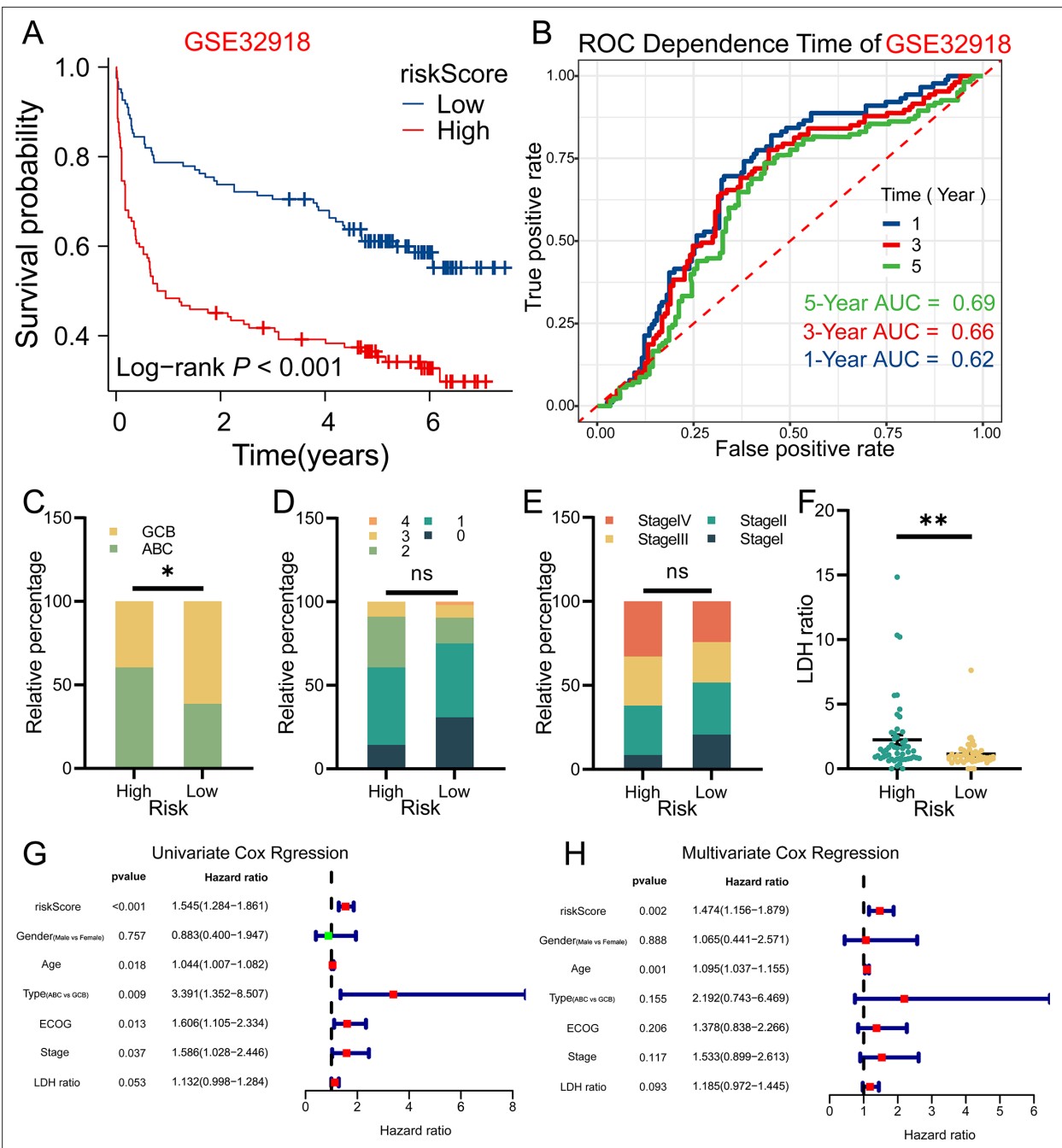

**Figure 5.** The riskScore of three tumor purity-related genes (TPGs) signature prognostic model was an independent prognostic factor in diffuse large B-cell lymphoma (DLBCL) patients. (**A**) Survival analysis results in GSE32918 dataset was consistent to that of GSE53786 dataset (n $_{Low}$ = 124, n $_{High}$ = 125). (**B**) The prognostic model also did well in GSE32918 dataset. (**C**) High-risk group in GSE53786 dataset contained more ABC-type DLBCL, while low-risk group contained more GCB-type DLBCL ($\chi^2$ test). (**D**) The ECOG performance of two groups (GSE53786 dataset) showed no statistical difference ($\chi^2$ test). (**E**) More patients in high-risk group were at Stage III or Stage IV, and less patients were at Stage I, compared with low-risk group, although no statistical significance was shown (GSE53786 dataset) ($\chi^2$ test). (**F**) High-risk group had higher LDH ratio (GSE53786 dataset) (Wilcoxon rank sum test, data are shown as mean ± SEM). (**G**) The riskScore was associated with poor prognosis of DLBCL patients in GSE53786 dataset. (**H**) The riskSocre was an independent prognostic factor for DLBCL patients. *p<0.05, **p<0.01, ns, not significant. ABC, activated B cell; GCB, germinal center B cell; ECOG, Eastern Cooperative Oncology Group; LDH, lactic dehydrogenase.

The online version of this article includes the following figure supplement(s) for figure 5:

**Figure supplement 1.** The correlation between tumor purity and prognosis and riskScore in GSE32918 dataset.

## The prognostic value of VCAN, CD3G, and C1QB were validated by IHC assay

With the purpose of the further validation of the prognostic value of VCAN, CD3G, and C1QB, we detected the expression of these genes in CHCAMS cohort by IHC. For VCAN, the patients were divided into high and low groups according to the cut-off of the H-score (275.42) provided by X-tile. The survival analysis showed that patients with high expression of VCAN had higher OS rate (*Figure 6A*, p=0.003). For CD3G, previous study revealed that it was a component of T cell receptor complex, for which it could be regarded as a marker of T cells (*Wang et al., 2008*). Therefore, we assessed the expression level of CD3G by counting the CD3G+ T cells ratio, and divided patients by the cut-off (2.5%). The survival analysis revealed that patients with high CD3G+ T cells infiltration showed favorable prognosis (*Figure 6B*, p<0.001). For C1QB, the patients in high expression group (cut-off=82.41) showed adverse prognosis (*Figure 6C*, p=0.015). Although the detection of protein level was not convenient to build a prognostic model for the difference of assessment methods and the lake of coefficient, these results were in accordance with those of GEO datasets, which successfully proved the prognostic value of VCAN, CD3G, and C1QB.

Given that these genes could potentially influence the tumor purity of DLBCL, we then analyzed the relationship between them and CD68 + macrophages, CD4 + T cells, and CD8 + T cells. As was shown in *Figure 6—figure supplement 1A*, CD68 + macrophages [(17.75±1.05) %] account for more ratio than CD4 + T cells [(0.68±0.20) %] and CD8 + T cells [(6.69±0.56) %] (p<0.001, Kruskal-Wallis Test and Dunn's Test). In the survival analysis of these three types of immune cells, we found that CD68 + macrophages, CD8 + T cells, and CD4 + T cells were associated with better prognosis (*Figure 6D–F*, p=0.029, p=0.002, p=0.053). And XCELL and QUANTISEQ algorithms revealed that M1 macrophages accounted for more proportion than M2 macrophages in GSE53786 and GSE32918 (*Figure 6—figure supplement 1L–O*), which was confirmed in the CHCAMS cohort (*Figure 6—figure supplement 1P*). Besides, the ratio of CD3G+ T cells was positively correlated with that of CD68 + macrophages, CD8 + T cells, and CD4 + T cells, C1QB expression level was positively correlated with CD8 + T cells, and VCAN expression level was positively correlated with CD8 + T cells ratio (*Figure 6G*). GSEA analysis based on the differentially expressed genes between high-risk and low-risk group in the GEO datasets above revealed that the cellular adhesion, extracellular structures, and immune-related processes could result in the different outcome (*Figure 6H–I*).

In addition to the above analyses, we also explored the relationship between these three genes and location of DLBCL. It turned out that CD3G+ T cells ratio was higher in DLBCL originated from groin and testis, and VCAN featured higher expression in lymph node-originated DLBCL (*Figure 6J–K*, p<0.05, p<0.01, *Figure 6—figure supplement 1B–K*).

These results showed that VCAN, CD3G, and C1QB played important roles in the microenvironment of DLBCL, possibly regulating the immune infiltration via modulating the extracellular organization and cellular interaction.

## The TPGs signature model could also predict the drug sensitivity of DLBCL patients

In order to learn about the ability of the previously mentioned model to predict drug sensitivity, we performed the prediction with 'oncoPredict' package in R. Fifteen drugs (*Supplementary file 2*) included in the GDSC and used in clinical practice or under clinical trials (searched on Pubmed) were enrolled in this prediction analysis.

As is shown in *Figure 7A* (prediction of GSE53786), patients in high-risk group could be sensitive to Carmustine, Cytarabine, Oxaliplatin, Vincristine, Vorinostat, and Bortezomib, but no drug could work better in low-risk group. And in GSE32918 (*Figure 7B*), Carmustine, Cytarabine, Oxaliplatin, Vorinostat, Afuresertib, Bortezomib, Ibrutinib, and Tamoxifen could work better in high-risk group, and Vincristine (sensitivity score: low-risk vs high-risk=0.219±0.026 vs 0.223±0.031) could work better in low-risk group. The discrepancy between the prediction in two datasets might be due to the samples and sequencing platforms. However, the intersection analysis of the drugs to which the high-risk patients in both datasets could be sensitive revealed that Carmustine, Cytarabine, Oxaliplatin, Vorinostat, and Bortezomib could be reliable candidates for treating high-risk patients based on the three TPGs signature prognostic model (*Supplementary file 2*).

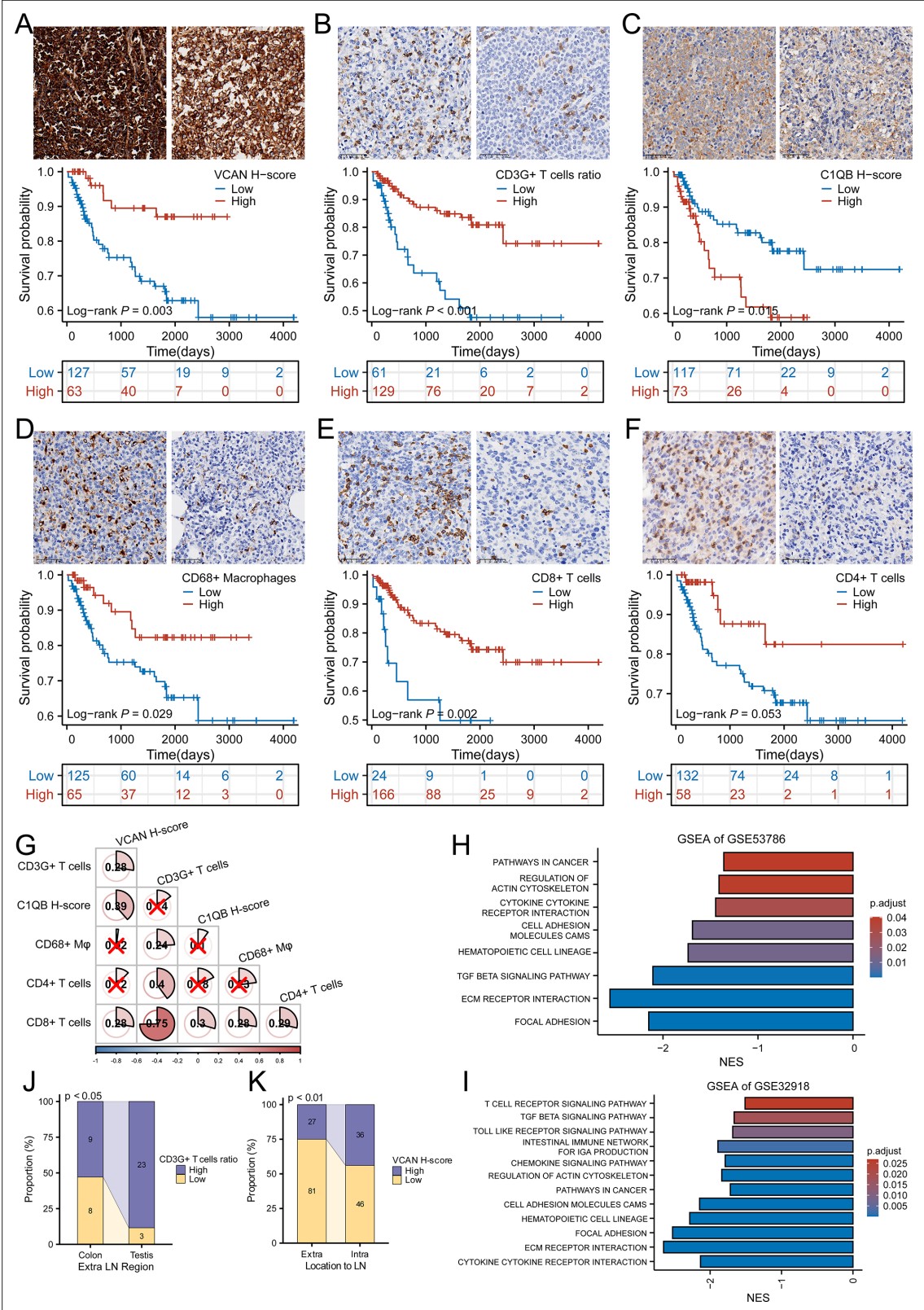

**Figure 6.** The analysis of CHCAMS cohort. (**A**) The representative image of VCAN staining of high and low expression groups and the survival analysis based on VCAN expression (n $_{Low}$ = 127, n $_{High}$ = 63). (**B**) The representative image of CD3G staining of high and low CD3G+ T cells ratio groups and the survival analysis based on CD3G+ T cells ratio (n $_{Low}$ = 61, n $_{High}$ = 129). (**C**) The representative image of C1QB staining of high and low expression groups and the survival analysis based on C1QB expression (n $_{Low}$ = 117, n $_{High}$ = 73). (**D**) The representative image of CD68 staining of high and low CD68 +

*Figure 6 continued on next page*

*Figure 6 continued*

macrophages ratio groups and the survival analysis based on CD68 + macrophages ratio (n $_{Low}$ = 125, n $_{High}$ = 65). (**E**) The representative image of CD8 staining of high and low CD8 + T cells ratio groups and the survival analysis based on CD8 + T cells ratio (n $_{Low}$ = 24, n $_{High}$ = 166). (**F**) The representative image of CD4 staining of high and low CD4 + T cells ratio groups and the survival analysis based on CD4 + T cells ratio (n $_{Low}$ = 132, n $_{High}$ = 58). (**G**) The correlation between VCAN, CD3G+ T cells ratio, C1QB and CD68 + macrophages, CD8 + T cells and CD4 + T cells ratio . '×' means no statistical significance. (**H-I**) GSEA analysis based on the differentially expressed genes between high-risk and low-risk group in GSE53786 and GSE32918. (**J**) The CD3G+ T cells infiltration varied from colon to testis originating diffuse large B-cell lymphoma (DLBCL) in male (Fisher's exact test). (**K**) The VCAN expression level was different between intra- and extra-lymph node DLBCL ($\chi^2$ test).

The online version of this article includes the following figure supplement(s) for figure 6:

**Figure supplement 1.** The immunoenvironment analysis and clinicopathological analysis of CHCAMS cohort and gene expression omnibus (GEO) datasets.

## Discussion

In this study, bioinformatics techniques were employed to identify three genes (VCAN, CD3G, C1QB) that exhibit associations with prognosis in both immune and stromal environments, thereby revealing their relationship with the prognosis of DLBCL patients. The findings indicate that higher expression of VCAN, increased infiltration of CD3G+ T cells, and decreased expression of C1QB are correlated with favorable prognostic outcomes. Conversely, a lower infiltration of CD68 + macrophages and lower infiltration of CD8 + T cells are associated with poorer prognosis. Furthermore, we investigated the relationship between risk genes related to tumor purity and treatment sensitivity and established a list of possible drugs that might be helpful for enhancing outcomes.

Previous studies have extensively investigated the VCAN gene in relation to tumorigenesis and metastasis (*Baghy et al., 2016*). VCAN, also known as versican, is a crucial component of extracellular matrix (*Wight, 2017*), and exists in several isoforms (*Fujii et al., 2015*). Research has shown that VCAN plays a multifaceted role in TME depending on the cell type expressing it. When expressed by myeloid cells, VCAN induces an anti-inflammatory and immunosuppressive microenvironment. Conversely, its expression by stromal cells typically leads to a pro-inflammatory response (*Wight et al., 2020*). In gastric cancer, high expression of VCAN has been associated with increased infiltration of fibroblasts, significant enrichment of stromal-associated signaling pathways, and poor prognosis (*Song et al., 2022*). In hepatocellular carcinoma, VCAN exhibits a strong association with immune checkpoint gene expression (*Wang et al., 2022*). Despite these findings in other tumor types, the role of VCAN in DLBCL has not been explored yet. Our study reveals that high expression of VCAN is actually associated with a more favorable prognosis. This suggests that VCAN may have different functions in different tumor types. One possible mechanism through which VCAN influences prognosis is that VCAN overexpression in DLBCL may also impact tumor cell proliferation. A study has shown that overexpression of VCAN V1 has an inhibitory effect on cell proliferation, partly due to its promotion of activation-induced cell death in lymphoid cell lines (*Fujii et al., 2015*). Hence, the high expression of VCAN in DLBCL could impact not only the TME but also tumor cell proliferation, suggesting a potential mechanism for the observed preferable prognosis.

C1q is synthesized in the tumor microenvironment and functions as an extracellular matrix protein, and C1QB is a component of C1q (*Yang et al., 2022*). Previous studies have provided insights into the diverse roles of C1q in cancer progression. However, the majority of these results, as observed in non-small cell lung carcinoma and gastric cancer, indicate that high C1q expression in TME is associated with a poor prognosis (*Li et al., 2023*; *Mangogna et al., 2019*; *Jiang et al., 2020*). Additionally, C1QB has been found to exert an impact on the TME and is positively associated with infiltration levels of CD8 + T cell, as well as with M1 and M2 macrophages in osteosarcoma (*Chen et al., 2021a*). Moreover, C1QB expression shows a positive correlation with predictive biomarkers for immunotherapy, such as PD-L1 expression and CD8 + T cell infiltration (*Jiang et al., 2020*). Furthermore, in malignant melanoma, C1QB promotes proliferation, migration, and invasion, while inhibiting cell apoptosis (*Zheng et al., 2021*), and the high-expression group exhibits significant enrichment of genes related to immune and apoptosis (*Yang et al., 2022*). In our study, we found that high expression of C1QB in DLBCL was associated with a worse prognosis and positively correlated with CD8 + T cells infiltration. Based on these findings, we propose that C1QB in DLBCL might share similarities with its functions in other tumor types, particularly regarding the promotion of recruitment and subsequent deactivation of CD8 + T cells within the TME through the induction of immune checkpoint effects. These results

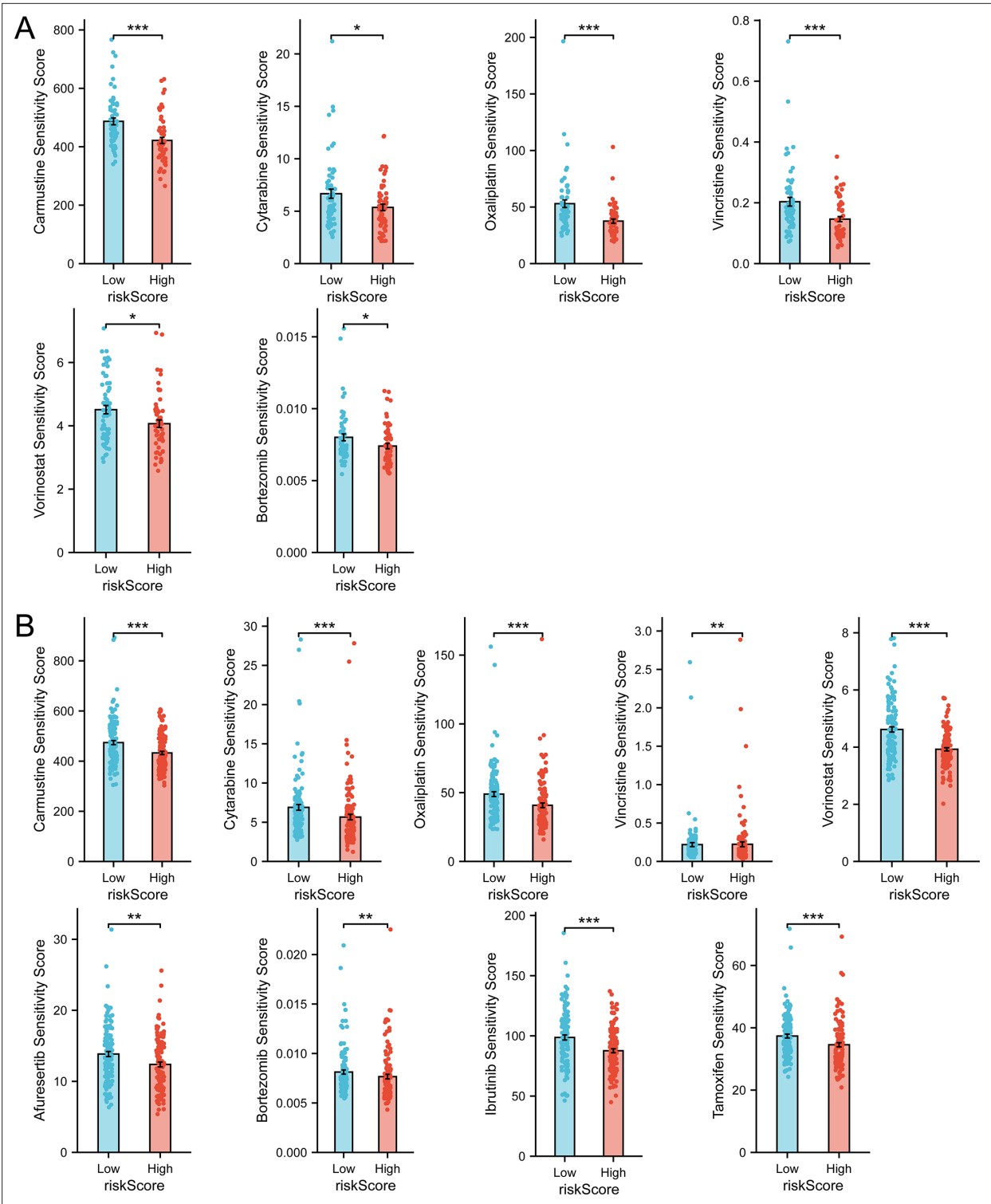

**Figure 7.** Drug sensitivity prediction revealed therapeutic candidates for high-risk group. (**A**) Drug sensitivity prediction results with statistical significance in GSE53786 dataset (n $_{Low}$ = 60, n $_{High}$ = 59, Wilcoxon rank sum test, data are shown as mean ± SEM ). (**B**) Drug sensitivity prediction results with statistical significance in GSE32918 dataset (n $_{Low}$ = 124, n $_{High}$ = 125, Wilcoxon rank sum test, data are shown as mean ± SEM). According to 'oncoPredict' algorithm, sensitivity score indicates IC50 of drugs, with higher sensitivity score indicating lower sensitivity.

shed light on the intricate role of C1QB in TME and its potential significance as a prognostic marker in DLBCL.

CD3G is a member of the TCR/CD3 complex primarily expressed in lymphocytes subgroups. It plays a crucial role in initiating the activation of T cells (*Wang et al., 2008*). It is also involved in coupling antigen recognition (*Chen et al., 2021b*). It is reported to be associated with long-term OS and good prognosis in breast invasive carcinoma (*Wang et al., 2019*) as well as in head and neck squamous cell carcinoma (*Wang et al., 2021*). However, its role in DLBCL has not been fully explored. In our study, we revealed that high infiltration of CD3G+ T cells is correlated with good prognosis. The infiltration of CD3G+ T cells was found to be positively related to the infiltration of CD8+, CD4 +, and CD68+ cells. This indicates that CD3G+ T cells in DLBCL may enhance the tumor antigen recognition process and stimulate the infiltration of immune cells, leading to an increased abundance of immune cell infiltration in the TME. The presence of CD3G+ T cells in the TME may contribute to a favorable prognosis by facilitating the activation of immune responses against tumor cells.

Macrophages play a crucial role in TME, and CD68 is a surface marker specific to macrophages. Macrophages can be roughly classified into two types based on their functional features: M1 or M2. M1 macrophages exert anti-tumor effects, whereas M2 macrophages promote tumor growth and progression in TME (*Zhang et al., 2021*). A previous study found that low infiltration of CD68 + macrophages was associated with an inferior prognosis (*Croci et al., 2021*). Similarly, our study has yielded similar results, revealing a noteworthy correlation between a high proportion of CD68 + macrophages in the TME and improved prognosis. Additionally, by analyzing the datasets, we observed a higher proportion of M1 macrophages infiltration compared to M2 macrophages. This suggests that, within our DLBCL cohort, these macrophages may also exhibit the M1 phenotype and consequently play a protective role against tumor progression.

CD8 is widely recognized as a marker of CD8 + T cells, also known as cytotoxic T cells (*Farhood et al., 2019*). These cells are crucial for the immune response against tumors. However, in DLBCL, CD8 + T cells exhibits elevated levels of inhibitory molecules on their surface, such as PD-1, PD-L1, and TIM3. High expression of TIM3, an inhibitory immune checkpoint receptor, on CD8 + T cells has been associated with tumor progression and poor outcomes (*Roussel et al., 2021*; *Xu-Monette et al., 2019*). These inhibitory molecules may impair the function of CD8 + T cells and hinder their anti-tumor activity. Surprisingly, our study demonstrates a correlation between the infiltration of CD8 +T cells and favorable prognosis in DLBCL. Here, we propose a hypothesis that in our study, the observed high expression of VCAN might create a suppressive environment for PD-1 + CD8+ T cells (*Wight et al., 2020*; *Hirani et al., 2021*). Intriguingly, our study revealed a statistically significant correlation between VCAN expression, C1QB expression, and CD8 + T cell infiltration. VCAN has the potential to modulate immune infiltration by reducing the immunosuppressive phenotype of immune cells (*Huang et al., 2021*), thus enabling a more efficient anti-tumor response. This aspect is still worth of consideration.

Taken together, our findings underscore the significant roles of VCAN, CD3G, and C1QB, which influence both the TME and the behavior of tumor cells. The interaction between each component and the TME is rather complicated. To fully comprehend the underlying mechanisms and identify potential therapeutic targets in DLBCL, further investigation is required.

However, this study still has several limitations that should be addressed. First, the patients included in this study were from a single center, which may introduce biases into the results. Although we made efforts to minimize these biases, it is inevitable that some may persist. Second, we hypothesized that VCAN, CD3G, and C1QB could serve as continuous prognostic parameters, thereby eliminating the need for a cut-off. However, the methodology used in this study, which utilized IHC staining to assess the protein expression levels, may have potential limitations. While IHC is a widely used technique, additional validation is needed to confirm the prognostic value of VCAN, CD3G, and C1QB in DLBCL. Furthermore, due to the potential variability in interpreting IHC results across different centers, a standardized coefficient and formula have not been established to calculate the final prognostic index for patients with DLBCL. Developing a standardized approach would be beneficial in ensuring consistent and accurate interpretation of IHC results. To address these limitations and expand upon our findings, future studies should strive to incorporate a diverse range of patients from multiple centers. Additionally, it is crucial to employ rigorous experimental techniques to authenticate the prognostic significance of VCAN, CD3G, and C1QB in DLBCL.

# Additional information

## Funding

| Funder | Grant reference number | Author |
|---|---|---|
| Shenzhen High-level Hospital Construction Fund | | Wenting Huang |
| CAMS Innovation Fund for Medical Sciences | 2022-I2M-C&T-B-062 | Liming Wang |

The funders had no role in study design, data collection and interpretation, or the decision to submit the work for publication.

## Author contributions

Zhenbang Ye, Conceptualization, Data curation, Formal analysis, Investigation, Visualization, Methodology, Writing – original draft; Ning Huang, Conceptualization, Data curation, Formal analysis, Validation, Visualization, Methodology, Writing – original draft; Yongliang Fu, Resources, Writing – review and editing; Rongle Tian, Writing – review and editing; Liming Wang, Resources, Formal analysis, Supervision, Methodology; Wenting Huang, Conceptualization, Resources, Supervision, Methodology, Writing – review and editing

## Author ORCIDs

Zhenbang Ye ⓘ https://orcid.org/0000-0002-3169-5023
Ning Huang ⓘ http://orcid.org/0000-0002-8874-2601
Liming Wang ⓘ http://orcid.org/0000-0002-0418-405X

## Ethics

The study was designed according to the Declaration of Helsinki and approved by the institutional ethics committee of Cancer Hospital Chinese Academy of Medical Sciences. Informed consent was taken from all the patients. (Reference number: 2020-20).

Reviewer #2 (Public Review): https://doi.org/10.7554/eLife.92841.3.sa1
Author response https://doi.org/10.7554/eLife.92841.3.sa2

# Additional files

## Supplementary files

- Supplementary file 1. The clinicopathological characteristics of CHCAMS cohort.
- Supplementary file 2. The prediction of drug sensitivity.
- Supplementary file 3. The original data of macrophage subtype ratio of CHCAMS cohort.
- Supplementary file 4. The original data of CHCAMS cohort.
- MDAR checklist

## Data availability

Data from GEO are GSE53786 and GSE32918. Original human subjects data of CHCAMS cohort are provided as supplementary files.

The following previously published datasets were used:

| Author(s) | Year | Dataset title | Dataset URL | Database and Identifier |
|---|---|---|---|---|
| Scott DW, Wright GW, Williams PM, Lih CJ | 2014 | DLBCL cell-of-origin by gene expression in FFPET | https://www.ncbi.nlm.nih.gov/geo/query/acc.cgi?acc=GSE53786 | NCBI Gene Expression Omnibus, GSE53786 |

*Continued on next page*

*Continued*

| Author(s) | Year | Dataset title | Dataset URL | Database and Identifier |
|---|---|---|---|---|
| Barrans SL, Care MA | 2012 | Whole genome expression profiling based on paraffin embedded tissue can be used to classify diffuse large b-cell lymphoma and predict clinical outcome | https://www.ncbi.nlm.nih.gov/geo/query/acc.cgi?acc=GSE32918 | NCBI Gene Expression Omnibus, GSE32918 |

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
