## [Editor Report · eLife assessment]

This study presents a predictive scoring system in DLBCL based on the expression of three tumour microenvironment-related genes. Such a scoring system seems **useful** for predicting tumour purity levels in DLBCL. The provided evidence showing an association between worse DLBLC prognosis and high-risk score is **solid**, but it is **incomplete** to draw a clear conclusion about the links between risk score and drug sensitivity.

---

## [Referee Report · Reviewer #2 (Public Review)]

In this study, Zhenbang Ye and colleagues investigate the links between microenvironment signatures, gene expression profiles, and prognosis in diffuse large B-cell lymphoma (DLBCL). They show that increased tumor purity (ie, a higher proportion of tumor cells relative to surrounding stromal components) is associated with worse prognosis. They then show that three genes associated with tumor purity (VCAN, CD3G, and C1QB) correlate with patterns of immune cell infiltration and can be used to create a risk scoring system that predicts prognosis, which can be replicated by immunohistochemistry (IHC), and response to some therapies.

(1) The two strengths of the study are its relatively large sample size (n = 190) and the strong prognostic significance of the risk scoring system. It is worth noting that the validation of this scoring with IHC, a simple technique already routinely used for the diagnosis and classification of DLBCL, increases the potential for clinical translation. However, the correlative nature of the study limits the conclusions that can be drawn in regards to links between the risk scoring system, the tumor microenvironment, and the biology of DLBCL.

(2) The tumor microenvironment has been extensively studied in DLBCL and a prognostic implication has already been established (for instance, Steen et al., Cancer Cell, 2021). In addition, associations have already been established in non-Hodgkin lymphoma between prognosis and expression of C1QB (Rapier-Sharman et al., Journal of Bioinformatics and Systems Biology, 2022), VCAN (S. Hu et al., Blood, 2013), and CD3G (Chen et al., Medical Oncology, 2022). Nevertheless, one of the strengths and novelty aspect of the study is the combination of these 3 genes into a risk score that is also valid by immunohistochemistry (IHC), which substantially facilitates a potential clinical translation.

(3) Figures 1A-B: tumor purity is calculated using the ESTIMATE (Estimation of Stromal and Immune cells in Malignant Tumor tissues using Expression data) algorithm (Yoshihara et al., Nature Communications, 2013). The ESTIMATE algorithm is based on two gene signatures ("stromal" and "immune"). It is therefore expected that tumor purity measured by the ESTIMATE algorithm will correlate with the expression of multiple genes. Importantly, C1QB is included in the stromal signature of the ESTIMATE algorithm meaning that, by definition, it will be correlated with tumor purity in that setting.

(4) Figure 2A: as established in figure 1C, high tumor purity is associated with worse prognosis. Later in the manuscript, it is also shown that C1QB expression is associated with worse prognosis. However, figure 2A shows that C1QB is associated with decreased tumor purity. It therefore makes it less likely that the prognostic role of C1QB expression is related to its impact on tumor purity. The prognostic impact could be related to different patterns of immune cell infiltration, as shown later. However, the evidence presented in the study is correlative and nature and not sufficient to draw this conclusion.

(5) Figure 3G: although there is a strong prognostic implication of the risk score on prognosis, the correlation between the risk score and tumor purity is significant but not very strong (R = 0.376). It is therefore likely that other important biological factors explain the correlation between the risk score and prognosis, as suggested in the gene set enrichment analysis that is later performed.

(6) Figure 6: the drug sensitivity analysis includes a wide range of established and investigational drugs with varied mechanisms of action. Although the difference in sensitivity between tumors with low and high risk scores show statistical significance for certain drugs, the absolute difference appears small in most cases and is of unclear biological significance. In addition, even though the risk score is statistically related to drug sensitivity, there is no direct evidence that the differences in drug sensitivity are directly related to tumor purity.

---

## [Author Response]

The following is the authors’ response to the original reviews.

**eLife assessment**
The findings in this study are useful and may have practical implications for predicting DLBCL risk subject to further validating the bioinformatics outcomes. We found the approach and data analysis solid. However, some concerns regarding the drug sensitivity prediction and the links between the selected genes for the risk scores have been raised that need to be addressed by further functional works.

Thanks for your high recognition for our study. In fact, we have searched the treatment information of DLBCL patients in our own cohort, however, unfortunately all patients were treated strictly according to the guidelines issued by authorities of China, which suit Chinese patients fine but do not include the drugs explored in the present study. Therefore, more further investigations should be designed and conducted to validate our conclusion. Here, we provided a possible direction for future studies base on large cohorts, which could not only provide more reliable conclusions, but gain more attentions to the role of tumor microenvironment in influencing outcome and drug sensitivity.

**Public Reviews:**

Sincere thanks for all reviewers’ positive comments on our study and their helpful recommendations for improving our manuscript. For this part, we have sorted out the comments and recommendations from all reviewers, and made corresponding revisions. And here are our responses.

(1) How did we determined the three genes (VCAN, C1QB and CD3G) in the prognostic model?

Just as was mentioned in the “Prognostic model” in Materials and Methods section, the gene was selected by “survival” package in R. After we obtained the nine genes, we input the expression value of them, and analyzed with “survival” package in R. And the function “step” in that package can optimize the model, that is, to construct a model with as less factors as possible, and the finally enrolled factors were representative and presented the least collinearity. Through this way, the prognostic model we got could be more practical in clinical practice.

(2) Different centers have different protocols of IHC, so how could we put this model into clinical practice under this circumstance?

Not only did different centers have different protocols, the materials like antibodies also vary. Therefore, there is actually a long way to go in putting our study into clinical practice. As far as we’re concerned, there are at least three problems to solve. First, diagnostic antibodies should be used in clinical practice, which usually manifest better specificity and sensitivity. And this may be the reason why the staining of VCAN and C1QB was strong and difficult to differentiate. Second, a standardized protocol should be made. Last but not least, more precise analyses and studies should be conducted to make it clear which type of cells specifically express these genes (just as was mentioned by Reviewer #2). We are now endeavoring to solve these problems by utilizing as many techniques as possible, like multi-omics and mIHC. From revealing the true expression pattern to developing high quality antibodies and even standardized test kit, we are looking forward to a clinical translation.

(3) The analyses about immune infiltration and the key genes in DLBCL were superficial, limited within the correlation analyses.

Due to the model constructed based on tumor purity of DLBCL, the risk score could be associated with the enrichment of cell functions. We conducted GSEA analysis based on the differentially expressed genes between high-risk group and low-risk group in the two datasets (Figure 5H-I). It showed that the extracellular organization and cellular adhesion were different between the two groups, in which way the immune infiltration and activity might be regulated owing to the motility of immune cells. Besides, we have validated the infiltration of M1 macrophages and M2 macrophages with our own cohort (Supplementary Figure 3P).

(4) The drug sensitivity was just analyzed based on the model, which should be validated in real world research or lab study. And the sensitivity score seemed not different too much in most cases, even though there were statistical significance.

We tried to search the treatment information of DLBCL patients in our own cohort, however, unfortunately all patients were treated strictly according to the guidelines issued by authorities of China, which suit Chinese patients fine but do not include the drugs explored in the present study. Therefore, more further investigations should be designed and conducted to validate our conclusion. Here, we provided a possible direction for future studies base on large cohorts, which could not only provide more reliable conclusions, but gain more attentions to the role of tumor microenvironment in influencing outcome and drug sensitivity. As for the differences between high- and low-risk group, as a matter of fact, sometimes a little dose of drug could have a huge effect, because the dose-effect curve is usually nonlinear. Therefore, reduce the dose, even just 1%, the adverse effects could be avoided. To sum up, the drug sensitivity analyses in our study could provide more possibility for clinical trial and practice, and we are taking it into consideration to design reasonable clinical research.

(5) C1QB was associated with decreased tumor purity and worse prognosis, but decreased tumor purity was related to better prognosis. How to elucidate the contradiction?

Just as discussed in Discussion section, previous studies have revealed the role of C1QB in promoting an immunosuppressive microenvironment in cancer (see reference 22-26). C1QB might recruit the infiltration of pro-tumor immune cells, resulting in a reducing tumor purity on its perspective. However, the immune microenvironment was regulated by multi factors which form a network and combat or synergize each other. The statistical analysis often gives a possible phenomenon, but could not provide mechanism explanation. Therefore, more mechanic studies are needed to reveal the connection and key node. This is exactly what we will explore next.

(6) Others:

(1) Line 51 has been rewritten.

(2) References for ESTIMATE algorithm (reference 16) and CD3G+ T cells has been added (reference 17).

(3) The illegible figure labels might be caused by the incompatibility between the PDF file we submitted and the submission system. We have provided the TIFF images in this revision, and the EPS file could be submitted to editors upon their requests.

(4) A supplement description has been added to the Figure legend of Figure 6 to make it clear.

(5) In order to explore the expression of key genes among different locations of DLBCL we performed analyses in Figure5 and supplementary Figure3. These results might be thought-provoking that the tumor microenvironment differs among DLBCLs even though they share similar histological characteristics.